# Prevalence and Molecular Characteristics of Polymyxin-Resistant *Pseudomonas aeruginosa* in a Chinese Tertiary Teaching Hospital

**DOI:** 10.3390/antibiotics11060799

**Published:** 2022-06-14

**Authors:** Chenlu Xiao, Yan Zhu, Zhitao Yang, Dake Shi, Yuxing Ni, Li Hua, Jian Li

**Affiliations:** 1Department of Laboratory Medicine, Ruijin Hospital Affiliated to Shanghai Jiao Tong University School of Medicine, Shanghai 200025, China; chenlu_xiao@hotmail.com; 2Department of Clinical Microbiology, Ruijin Hospital Affiliated to Shanghai Jiao Tong University School of Medicine, Shanghai 200025, China; 3Monash Biomedicine Discovery Institute, Infection Program and Department of Microbiology, Monash University, Melbourne 3800, Australia; yan.zhu@monash.edu; 4Department of Emergency, Ruijin Hospital Affiliated to Shanghai Jiao Tong University School of Medicine, Shanghai 200025, China; yangzhitao@hotmail.fr; 5Department of Infection Control, Ruijin Hospital Affiliated to Shanghai Jiao Tong University School of Medicine, Shanghai 200025, China; dake_shi@126.com (D.S.); yuxing_ni@126.com (Y.N.); 6School of Public Health, Shanghai Jiao Tong University School of Medicine, Shanghai 200025, China

**Keywords:** *Pseudomonas aeruginosa*, polymyxin resistance, phenotypes, genomics

## Abstract

Polymyxin-resistant *Pseudomonas aeruginosa* is a major threat to public health globally. We investigated the prevalence of polymyxin-resistant *P. aeruginosa* in a Chinese teaching hospital and determined the genetic and drug-resistant phenotypes of the resistant isolates. *P. aeruginosa* isolates identified by MALDI-TOF MS were collected across a 3-month period in Ruijin Hospital. Antimicrobial susceptibility was determined by a Vitek-2 Compact system with broth dilution used to determine polymyxin B (PMB) susceptibility. Polymyxin-resistant isolates were further characterized by molecular typing using PCR, multi-locus sequence typing (MLST) and whole-genome sequencing. Phylogenetic relationships were analyzed using single nucleotide polymorphism (SNP) from the whole-genome sequencing. Of 362 *P. aeruginosa* isolates collected, 8 (2.2%) isolates from separate patients across six wards were polymyxin-resistant (MIC range, PMB 4–16 μg/mL and colistin 4–≥16 μg/mL). Four patients received PMB treatments (intravenous, aerosolized and/or topical) and all patients survived to discharge. All polymyxin-resistant isolates were genetically related and were assigned to five different clades (Isolate 150 and Isolate 211 being the same ST823 type). Genetic variations V51I, Y345H, G68S and R155H in *pmrB* and L71R in *pmrA* were identified, which might confer polymyxin resistance in these isolates. Six of the polymyxin-resistant isolates showed reduced susceptibility to imipenem and meropenem (MIC range ≥ 16 μg/mL), while two of the eight isolates were resistant to ceftazidime. We revealed a low prevalence of polymyxin-resistant *P. aeruginosa* in a Chinese teaching hospital with most polymyxin-resistant isolates being multidrug-resistant. Therefore, effective infection control measures are urgently needed to prevent further spread of resistance to the last-line polymyxins.

## 1. Introduction

*Pseudomonas aeruginosa* is an opportunistic Gram-negative pathogen commonly found in soil, water and plants [1]. This versatile pathogen is associated with a variety of nosocomial and community-acquired infections affecting skin and soft tissue, bone and joints, the urinary tract and, importantly, the bloodstream and respiratory tract (including ventilator-associated pneumoniae (VAP)) [2]. *P. aeruginosa* infections often have higher rates of morbidity and mortality and higher treatment costs when compared with other bacterial pathogens, especially in patients with chronic diseases or compromised immune systems [1]. Although the carbapenems were introduced to treat serious infections caused by organisms such as multidrug-resistant (MDR) *P. aeruginosa*, resistance to the carbapenems has increased globally with between 10 and 50% of *P. aeruginosa* isolates in most countries now considered to be carbapenem-resistant [3]. In 2021, it was estimated that approximately 18.9–23.0% of *P. aeruginosa* isolates in China were carbapenem-resistant (http://www.chinets.com/, accessed on 5 January 2022). *P. aeruginosa* is included among a group of MDR pathogens (*Enterococcus faecium*, *Staphylococcus aureus*, *Klebsiella pneumoniae*, *Acinetobacter baumannii*, *P. aeruginosa* and *Enterobacter* species) which ‘ESKAPE’ the effects of commonly used antibiotics [4]. Indeed, carbapenem-resistant *P. aeruginosa* is listed as one of only three World Health Organization Priority 1 (Critical) pathogens urgently requiring the discovery and development of novel antibiotics [5].

The ‘old’ polymyxins (colistin and polymyxin B) were first introduced into the clinic in the late 1950s but subsequently abandoned due to nephrotoxicity concerns [6]. Given the ever-increasing resistance to other antibiotics including the carbapenems and aminoglycosides, polymyxins were re-introduced to clinical practice for the treatment of problematic Gram-negative ‘superbugs’ in the early 2000s [7]. The polymyxins remain an important last-line treatment as they retain excellent activity against many of these problematic pathogens [8]. For example, in 2021 the China Antimicrobial Surveillance Network (http://www.chinets.com/, accessed on 5 January 2022) estimated that only 2.0% of *P. aeruginosa* isolates and 1.0% of *Escherichia coli* and *K. pneumoniae* isolates in China were resistant to colistin and polymyxin B. In many countries, the polymyxins are the only accessible or affordable therapeutic option for carbapenem-resistant organisms [9]. Worryingly, resistance to polymyxins, including in *P. aeruginosa*, is increasingly reported both in humans and the surrounding environment [10,11,12]. As drug resistance genes such as extended-spectrum β-lactamases (ESBLs), metallo-β-lactamases (MBLs) and New Delhi metallo-β-lactamases (NDMs), which provide resistance to more recently developed antibiotics, continue to increase in prevalence globally [13,14], a concurrent increase in polymyxin resistance threatens the utility of one of the last available treatment options for MDR Gram-negative pathogens.

Resistance to the polymyxin is primarily chromosomally mediated and involves several different mechanisms, in particular lipopolysaccharide (LPS) modification [15]. Plasmid-mediated polymyxin-resistance via *mcr* genes has also recently been reported [16,17,18]. In China, most studies that have investigated mechanisms of polymyxin resistance have examined *Enterobacterales*, with few studies examining resistance in *P. aeruginosa* [19,20,21]. The present study aimed to investigate the prevalence, molecular characteristics and antibiotic susceptibility of polymyxin-resistant *P. aeruginosa* isolated from patients in a Chinese tertiary teaching hospital.

## 2. Methods and Materials

### 2.1. Bacterial Isolates and Antibiotic Susceptibility Testing

This study (No. 2022051) was approved by the Ruijin Hospital Ethics Committee (Shanghai Jiao Tong University School of Medicine, Shanghai, China). Ruijin Hospital is a 3624-bed tertiary care teaching hospital with approximately 130,000 hospital admissions per year. During August to October in 2021, all isolates of *P. aeruginosa* determined to be polymyxin resistant were collected by the Department of Clinical Microbiology. Isolates were identified by the MADLI-TOF MS system (BioMerieux, Missouri, France, software version 3.2) and antimicrobial susceptibility testing performed using the Vitek-2 AST-N335 Compact system (Bio Mérieux, France). The antimicrobial agents tested included: amikacin (aminoglycosides); aztreonam (monobactam); cefepime; cefoperazone/sulbactam; ceftazidime; ciprofloxacin and levofloxacin (quinolone); colistin; meropenem and imipenem (carbapenems); doxycycline; piperacillin/tazobactam; tigecycline (glycylcycline); ticarcillin/clavulanic acid; tobramycin and trimethoprim/sulfamethoxazole. MICs of polymyxin B and colistin were determined using broth microdilution with *E. coli* ATCC 25922 used as the quality control strain. All results (except polymyxins) were interpreted according to the Clinical and Laboratory Standards Institute (CLSI) guidelines [22].

### 2.2. Genome Sequencing, Assembly and Annotation

Genomic DNA from each polymyxin-resistant isolate was extracted using the cetyltrimethyl ammonium bromide (CTAB) method with minor modifications [23]. DNA quantity, quality and integrity were checked using a Qubit Flurometer (Invitrogen, Waltham, MA, USA) and NanoDrop Spectrophotometer (Thermo Scientific, Wilmington, DE, USA). Sequencing libraries were generated using the TruSeq DNA Sample Preparation Kit (Illumina, San Diego, CA, USA) and the Template Prep Kit (Pacific Biosciences, Melno Park, CA, USA), and genome sequencing performed by Shanghai Personal Biotechnology (Shanghai, China) on the Illumina Novaseq platform. Genome function elements prediction included prediction of coding genes. Gene prediction was performed by Glimmer 3.02 (http://ccb.jhu.edu/software/glimmer/index.shtml, accessed on 8 February 2022). CRISPRs were identified by the CRISPR recognition tool. Subsequently, the VFDB (Virulence Factors of Pathogenic Bacteria) and CARD (The Comprehensive Antibiotic Resistance) databases were used to retrieve pathogenicity genes and antibiotic resistance genes, respectively. De novo genome assembly was conducted using A5-Miseq (v20160825) and SPAdes (v3.12.0), followed by base correction using Pilon. Function annotation was completed by blast search against different databases including NR (Non-Redundant Protein Database), GO (Gene Ontology), KEGG (Kyoto Encyclopedia of Gene and Genomes) and COG (Cluster of Orthologous Groups of proteins) to give an overview of the genome information.

High-quality sequence reads were mapped to the *P. aeruginosa* strain reference genome B136-33 (GenBank accession no. CP004061.1) using BWA [24] (version 0.7.12-r1039). This genome was chosen because it most closely matched the genomes of the 3 polymyxin-resistant *P. aeruginosa* isolates identified. *P. aeruginosa* B136-33 belongs to ST 1024 and contains *pmrA*, *pmrB*, *phoP* and *bla*_oxa-50_ genes; this strain was also included in the phylogenetic analysis. The alignments were improved using the Picard package (http://sourceforge.net/projects/picard/, accessed on 9 February 2022) with two commands: “FixMateInformation” and “MarkDuplicates”. Where multiple read pairs had identical external coordinates, the pair with the best mapping quality was retained and the others marked as duplicates. A two-step local realignment of the mapped reads around indels was undertaken using the GATK package: firstly, suspicious intervals that likely needed realigning were determined by the “RealignerTargetCreator” command; then, realignment of such intervals was performed by the “IndelRealigner” command [25]. Subsequently, variant calling was carried out using the Bayesian approach which was implemented in the GATK package (https://software.broadinstitute.org/gatk/, accessed on 10 February 2022). The variants were further filtered based upon the following criteria: RMS mapping quality of ≥25, site quality score of ≥30, variant confidence/quality by depth of ≥2, ≥16 reads covering each site with 8 reads mapping to each strand, and the reads covering a major variant were at least five times greater than that of the minor variant. Sites that failed these criteria in any strain were removed from the analysis. The complete genome sequences of the polymyxin-resistant *P. aeruginosa* isolates were deposited in the NCBI BioProject repository under the accession number PRJNA846971.

### 2.3. MLST and Phylogenetic Analyses

Multiple housekeeping gene internal fragments were amplified by PCR, their sequences determined and the variations in the isolates analyzed. The PubMLST database for *P. aeruginosa* was employed (https://pubmlst.org/organisms/pseudomonas-aeruginosa, accessed on 22 February 2022) for multilocus sequence typing (MLST) using the following seven housekeeping genes: *ascA*, *aroE*, *guaA*, *mutL*, *nuoD*, *ppsA* and *trpE*. For isolates with genome sequences available, MLST was predicted using mlst (https://github.com/tseemann/mlst, accessed on 24 February 2022) [26].

Using the genome of model strain PAO1 as a reference, Snippy (https://github.com/tseemann/snippy, accessed on 24 February 2022) was employed to identify SNPs across 385 complete genomes of *P. aeruginosa* obtained from RefSeq database and the draft genomes of the 8 polymyxin-resistant isolates in the present study. Core SNPs were concatenated and aligned using snippy-multi script. Subsequently, a maximum likelihood tree was constructed based on core-SNP alignment using IQ-TREE 2 (using a general time-reversible model with ascertainment bias correction, 1000 bootstraps) and then visualized using ggtree with ‘daylight’ layout [27,28].

## 3. Results

### 3.1. Patient Demographics and Characteristics of the Polymyxin-Resistant P. aeruginosa Isolates

In total, 362 *P. aeruginosa* isolates were collected across the study period with eight polymyxin-resistant isolates collected from separate patients across six different wards (Table 1). Colistin MICs were 4 μg/mL (one isolate) or ≥16 μg/mL (seven isolates) while polymyxin B MICs ranged from 4 to 16 μg/mL (Table 2). The patients from which the polymyxin-resistant *P. aeruginosa* isolates were obtained ranging in age from 37 to 87 years with a majority being frail adults with multiple complicated comorbidities. Three patients received intravenous polymyxin B sulfate and all eight patients from which polymyxin-resistant *P. aeruginosa* was isolated were subsequently discharged. With reference to the PAO1 (polymyxin-susceptible) genome, we identified six, one and one genetic variations in *pmrA*, *pmrB* and *phoQ* of the polymyxin-resistant isolates, respectively (Table 3). Among them, V51I, Y345H, G68S and R155H in *pmrB* and L71R in *pmrA* were known as variations that potentially contribute to polymyxin resistance [29,30,31,32]. Five of the polymyxin-resistant *P. aeruginosa* isolates contained the oxacillinase gene *bla*_oxa-50_ and all had the beta-lactamase gene *bla*_PDC_. No *mcr*-genes were detected, showing that *mcr* genes were not the cause of polymyxin resistance. Isolates 150 and 211 belong to the same sequence type (ST823). While isolate 160 did not belong to any known sequence type, it was very similar to the ST823 sequence type based on the seven housekeeping genes.

### 3.2. Antimicrobial Susceptibility and Detection of Polymyxin-Resistant P. aeruginosa

Antibiotic susceptibilities are shown in Table 2. Six of the eight (75%) polymyxin-resistant *P. aeruginosa* isolates had reduced susceptibility to carbapenems and were resistant to both imipenem and meropenem (MIC range, 0.5–≥16 μg/mL). The lowest level of resistance was to ceftazidime (MIC range, 2–64 μg/mL), with resistance detected in only two of eight (25%) isolates.

The time at which polymyxin-resistant *P. aeruginosa* was detected in each patient during hospitalization relative to the commencement of treatment is shown in Figure 1. Although samples were collected from more than one location in all patients, polymyxin-resistance was only detected in one location per patient. For example, isolate 150 was detected in the wound of a burn patient, while polymyxin-susceptible *P. aeruginosa* was simultaneously detected in the blood of this patient. Of the four patients who received polymyxin treatment, polymyxin-resistant *P. aeruginosa* was detected prior to the commencement of treatment in one patient and after treatment commenced in the remaining three patients (as shown by the black arrows in Figure 1 which represent the commencement of polymyxin B treatment; the two arrows for isolate 211 represent commencement of topical treatment (first arrow) and IV plus aerosolized treatment (second arrow)).

### 3.3. Phylogenetic Analysis

Phylogenetic analysis of the GenBank complete genomes and the polymyxin-resistant genomes in the present study identified two distinct groups in *P. aeruginosa* (Figure 2). Group 1 is larger, and contains less virulent strain PAO1 and PAK, and also the cystic fibrosis strain LESB58 [33]. Isolates 190 and 207 from ST277 and isolate 149 from ST360 belong to this group and form a sub-lineage, whereas Group 2 tends to be smaller and includes the well-known virulent strain PA14 [34]. Five of the eight polymyxin-resistant isolates (150, 211, 167, 166 and 206) belong to this group, with 167, 150 and 211 being clustered into an individual sub-lineage, suggesting their close phylogenetic relationship. The most prominent sequence type in our hospital was ST823 (isolate 150 and 211) which belongs to one sub-lineage and has previously been reported primarily in Asia [35]. The 385 complete *P.aeruginosa* genomes are available in the Appendix A.

## 4. Discussion

Increasing polymyxin resistance threatens the utility of this important last-line class of antibiotics against otherwise untreatable Gram-negative bacteria. While there have been previous case reports of infections caused by polymyxin-resistant *P. aeruginosa* in China and other countries, these reports did not attempt to determine the prevalence of polymyxin-resistance in the associated institution [36,37,38]. Similarly, other studies have pooled data from multiple institutions to determine regional susceptibility to polymyxins [38,39]. Our hospital started using polymyxin B and colistin and their susceptibility testing in late 2020 for the treatment of carbapenem-resistant Gram-negative bacteria. In the present study, we analyzed the prevalence of polymyxin-resistant *P. aeruginosa* during 2021 and examined bacterial genetic and resistant phenotypes and the clinical characteristics of the patients from which they were isolated.

In our study from a single hospital, the prevalence over 2021 was 2.2% (8/362), which is consistent with the 2.0% average figure for polymyxins reported in 2021 by the China Antimicrobial Surveillance Network (http://www.chinets.com/, accessed on 5 January 2022). All eight polymyxin-resistant isolates have similar MICs of polymyxin B and colistin, which is not surprising given their structural similarity (differing only by a single amino acid) [12]. Of concern is that six of the eight isolates contained OXA, ODC and VIM lactamases which confer resistance to both imipenem and meropenem, although four of these isolates remained susceptible to ceftazidime. Using the definitions for MDR, extensively drug-resistant (XDR) and pandrug-resistant (PDR), isolate 190 was PDR and resistant to all antibiotics tested, while another five isolates (isolate 149, 150, 167, 207, 211) were XDR or MDR [40]. Only isolate 166 and 206 were not MDR. While the MDR *P. aeruginosa* ST most prevalent globally is ST235, none of the polymyxin-resistant *P. aeruginosa* isolates here matched this sequence type [41]. ST277 (isolate 190) has previously been reported in Brazil and was highly correlated with the accessory genome ST-274 (both in the same clade), the latter having been previously reported in Nepal [42,43]. While ST 277 has previously been reported in Brazil and contains 13 different antibiotic resistance genes, in our study isolate 190 contained only 5 antibiotic resistance genes [42] (Table 1). Isolate 206 in our study carried five antibiotic resistance genes and belongs to ST671 which has previously been reported in China [44]. All eight polymyxin-resistant isolates contained characteristic drug resistance genes and a genetic background confined to different geographic locations. Our finding is worrying given it severely limits the treatment options available.

Several studies have shown an association between polymyxin use and the emergence of polymyxin resistance in carbapenem-resistant *K*. *pneumoniae* in hospital, although results are controversial [45,46,47,48]. In our study, the eight polymyxin-resistant *P. aeruginosa* isolates were all obtained from patients with multiple comorbidities located in settings where infections commonly occur, necessitating long-term antibiotic therapy. Half of the patients from whom the resistant isolates were collected received polymyxin B treatment (either topical, IV or via inhalation) during hospitalization, while the remaining patients received no polymyxin therapy. While the small number of patients precludes determining any association between polymyxin use and the emergence of its resistance, we hope to collect more isolates to examine this relationship. Interestingly, polymyxin-resistant isolates were also isolated in the four patients who received no polymyxin therapy during the hospitalization, and further investigations are warranted.

The acquisition of polymyxin resistance is primarily due to mutations in two-component regulatory systems (TCSs), including PhoPQ and PmrAB [14,21,49]. Specific mutations trigger constitutive upregulation of the *pmrHFIJKLM-ugd* operon, which in turn leads to the covalent modification of lipid A phosphate groups with positively charged motifs such as phosphoethanolamine (pEtN) and/or 4-amino-4-deoxy-L-arabinose (L-Ara4N) [50,51,52]. Given an initial electrostatic interaction with lipid A of the Gram-negative outer membrane is a requirement for bacterial killing by the polymyxins, L-Ara4N and pEtN modifications decrease the negative charge of lipid A and consequently its interaction with the positively charged polymyxin molecules [53]. In *P. aeruginosa*, multiple TCSs (PmrAB, ParRS, CprRS, and ColRS) are involved with the addition of L-Ara4N to LPS and thus play a role in polymyxin resistance [54,55]. Recently, plasmid-mediated polymyxin resistance via the pEtN transferase *mcr* genes (*mcr*-1 to *mcr*-10) has also been reported, including in *P. aeruginosa* [56,57,58]. Furthermore, *P. aeruginosa* can readily acquire transmissible antibiotic resistance (AbR) genes, resulting in the emergence of MDR or XDR isolates [59]. In our study, all the polymyxin-resistant isolates had mutations in *pmrB*, a major TCS in Gram-negative bacteria which causes polymyxin resistance.

## 5. Conclusions

In conclusion, we reported a low prevalence of polymyxin-resistant *P. aeruginosa* soon after polymyxins were introduced in a Chinese tertiary teaching hospital. Given polymyxin-resistant *P. aeruginosa* is a major threat to public healthcare, it is important for clinical laboratories to detect polymyxin resistance and characterize the epidemiological trends in high-risk *P. aeruginosa* isolates to optimize the use of this last-line class of antibiotics. Furthermore, effective infection control measures are urgently needed to prevent further transmission of polymyxin resistance.

## Figures and Tables

**Figure 1 antibiotics-11-00799-f001:**
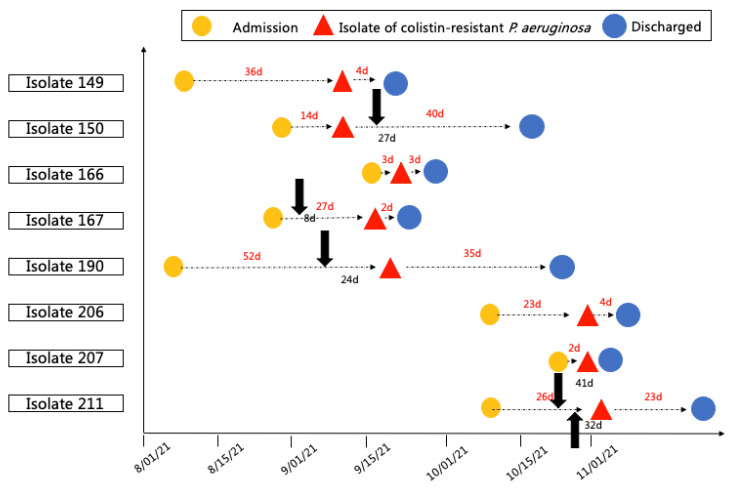
Timeline for the isolation of polymyxin-resistant *P. aeruginosa*. The red triangles show the time polymyxin-resistant *P. aeruginosa* was isolated. The black arrows show when polymyxin B treatment commenced.

**Figure 2 antibiotics-11-00799-f002:**
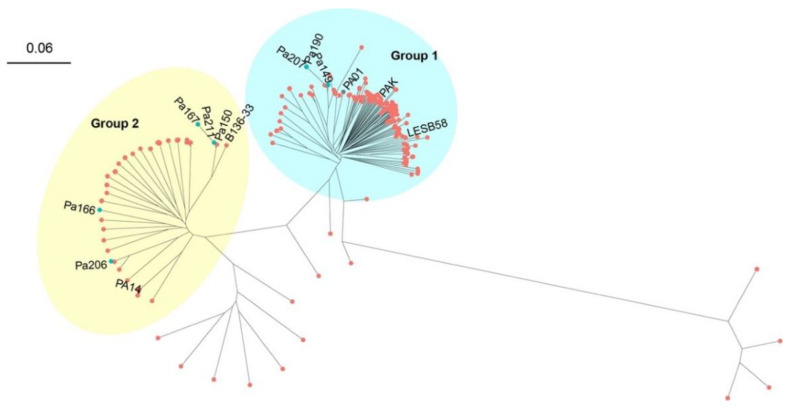
Phylogenetic tree of *P. aeruginosa* isolates constructed based on core-SNPs. Red nodes indicate the complete genomes obtained from the RefSeq database, whereas blue nodes indicate the genomes sequenced in this study.

**Table 1 antibiotics-11-00799-t001:** Patient demographics and main characteristics of the polymyxin-resistant *P. aeruginosa* isolates.

Isolate	Gender	Age (Year)	Underlying Disease	Ward	Polymyxin Treatment ^†^	Length of Hospital Stay (Day)	Outcome	Specimen (Collection Date, Day/Month/Year)	MLST Type	Carbapenem Resistance Gene
149	Female	37	Severe acute pancreatitis; sepsis	Emergency Intensive Care Unit (EICU)	No treatment; considered as colonization in bile	40	Survived	Bile (10 September 2021)	ST360	*bla* _OXA-50_ *, bla* _PDC-5_
150	Male	37	Skin, soft tissue and respiratory infection due to extensive burns (52% flame burns degree II-III)	Burns ward	Local use of topical polymyxin B plus IV polymyxin B sulfate 50 mg 12-hourly (both administered for the same 27 days)	54	Survived	Wound (10 September 2021)	ST823	*bla* _PDC-7_ *, bla* _VIM-2_
166	Female	70	Diffuse large B-cell lymphoma; neutropenia; decompensated cirrhosis due to autoimmune hepatitis.	Dermatology	No treatment; considered as colonization in sputum	6	Survived	Sputum (23 September 2021)	ST1621	*bla* _OXA-50_ *, bla* _PDC-10_
167	Female	48	Dermatomyositis; thrombocytopenia	Dermatology	Local use of topical polymyxin B for 8 days	29	Survived	Wound (23 September 2021)	NT (not detected)	*bla* _PDC-7_ *, bla* _VIM-2_
190	Male	51	Severe abdominal infection due to acute suppurative appendicitis with perforation; septic shock	EICU	IV polymyxin B sulfate 50 mg 12-hourly for 24 days	87	Survived	Extravasate Fluid (30 September 2021)	ST277	*bla* _OXA-50_ *, bla* _PDC-3_
206	Male	76	ANCA-associated vasculitis with cerebral infarction	Neurology	No treatment; considered as colonization in sputum	27	Survived	Sputum (3 November 2021)	ST671	*bla* _OXA-50_ *, bla* _PDC-10_
207	Male	61	Vitiligo	Respiratory	No treatment; considered as colonization in sputum	2	Survived	Sputum (3 November 2021)	ST277	*bla* _OXA-50_ *, bla* _PDC-5_
211	Male	47	Skin, soft tissue and respiratory infection due to extensive burns (75% flame burns, 50% degree III)	Burns wards	Local use of topical polymyxin B for 41 days; IV polymyxin B sulfate 50 mg 12-hourly plus aerosolized polymyxin B 25 mg 12-hourly (both administered for the same 32 days)	79	Survived	Sputum (4 November 2021)	ST823	*bla* _PDC-7_ *, bla* _VIM-2_

^†^ Topical treatment involved application of Funuo™ ointment containing 50,000 units of polymyxin B sulfate, 35,000 units of neomycin sulfate, 5000 units of bacitracin and 400 mg of lidocaine HCl per gram. IV, intravenous.

**Table 2 antibiotics-11-00799-t002:** Antimicrobial susceptibilities of the polymyxin-resistant *P. aeruginosa* isolates (MICs, μg/mL).

Isolate	CAZ	AZT	IMP	CIP	TIM	CPE	MEM	AK	CL	LEV	PMB	TM	TZP	SCF
149	16	32	≥16	1	≥128	16	≥16	4	4	0.5	4	≤1	≥128	16
150	16	≥64	≥16	≥4	≥128	8	≥16	≥64	≥16	≥8	4	≥16	32	≥64
166	2	8	1	2	16	2	0.5	≤2	≥16	0.5	4	≤1	≤4	≤8
167	16	4	≥16	≥4	≥128	8	≥16	≥64	≥16	≥8	16	≥16	16	≥64
190	≥64	≥64	≥16	≥4	≥128	≥16	≥16	≥64	≥16	≥8	8	≥16	≥128	≥64
206	2	4	2	≤0.25	32	4	≥0.25	≤2	≥16	0.25	4	≤1	8	≤8
207	≥64	≥64	≥16	≥4	≥128	16	≥16	≥64	≥16	≥8	16	≤1	≥128	≥64
211	16	4	≥16	≥4	≥128	16	≥16	≥64	≥16	≥8	4	≥16	32	≥64

CAZ, ceftazidime; AZT, aztreonam; IMP, imipenem; CIP, ciprofloxacin; TIM, ticarcillin/clavulanic acid; CPE, cefepime; MEM, meropenem; AK, amikacin; CL, colistin; LEV, levofloxacin; PMB polymyxin B; TM, tobramycin; TZP, piperacillin/tazobactam; SCF, cefoperazone/sulbactam.

**Table 3 antibiotics-11-00799-t003:** Genetic variations in *pmrA*, *pmrB* and *phoQ* of the polymyxin-resistant isolates.

Isolate	*pmrB*	*pmrA*	*phoQ*
149	1033T>C (Y345H)	212T>G (L71R)	-
150	43G>A (V15I), 202G>A (G68S), 1033T>C (Y345H)	-	-
166	43G>A (V15I), 202G>A (G68S), 1033T>C (Y345H)	212T>G (L71R)	-
167	464G>A (R155H), 1033T>C (Y345H)	212T>G (L71R)	
190	1033T>C (Y345H), 1105C>G (P369A), 1384G>A (A462T)	212T>G (L71R)	-
206	43G>A (V15I), 202G>A (G68S), 1033T>C (Y345H)	-	789G>T (Q263H)
207	1033T>C (Y345H), 1105C>G (P369A), 1384G>A (A462T)	212T>G (L71R)	-
211	43G>A (V15I), 202G>A (G68S), 1033T>C (Y345H)	-	-

## Data Availability

The data presented in this study are openly available in the NCBI BioProject repository under the accession number PRJNA846971.

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
