# Peer review of "Prevalence and Molecular Characteristics of Polymyxin-Resistant Pseudomonas aeruginosa in a Chinese Tertiary Teaching Hospital"

_antibiotics, 2022, doi:10.3390/antibiotics11060799_

Round 1
Reviewer 1 Report
The authors should do some minor English grammar changes in the manuscript
Author Response
Thank you for your letter and for the reviewers’ comments concerning our manuscript entitled “Prevalence and molecular characteristics of polymyxin-resistant Pseudomonas aeruginosa in a Chinese tertiary teaching hospital” (Submission ID: antibiotics-1748582). Those comments are all valuable and very helpful for revising and improving our paper, as well as the important guiding significance to our researches. We have studied comments carefully and have made correction which we hope meet with approval. The main corrections in the paper and the responds to the reviewer’s comments are as flowing:
All line and page numbers referred to below refer to the revised version of the manuscript.
Comments and suggestions for Authors:
The authors should do some minor English grammar changes in the manuscript.
Response: The manuscript has been edited throughout to improve the grammar and readability.
Other changes:
We tried our best to improve the manuscript and made some changes in the manuscript. These changes will not influence the content and framework of the paper.
We appreciate for Editors/Reviewers’ warm work earnestly, and hope that the correction will meet with approval.
Once again, thank you very much for your comments and suggestions.
Reviewer 2 Report
The authors claim that: "The present study aimed to investigate the prevalence, molecular characteristics, and antibiotic susceptibility of polymyxin-resistant P. aeruginosa isolated from patients in a Chinese tertiary teaching hospital. and that In China,many studies reported the mechanism of polymyxin...", so I have doubts about the novelty of the work. It should be clearly stated what is new about the work.MALDI-TOF MS was used in the study, but no details are reported, there is no chromatogram etc. Line 86-87 MODLI-TOF MS should be written as MALDI-TOF MS.
Author Response
Thank you for your letter and for the reviewers’ comments concerning our manuscript entitled “Prevalence and molecular characteristics of polymyxin-resistant Pseudomonas aeruginosa in a Chinese tertiary teaching hospital” (Submission ID: antibiotics-1748582). Those comments are all valuable and very helpful for revising and improving our paper, as well as the important guiding significance to our researches. We have studied comments carefully and have made correction which we hope meet with approval. The main corrections in the paper and the responds to the reviewer’s comments are as flowing:
All line and page numbers referred to below refer to the revised version of the manuscript. Please check the line numbers in the PDF file.
Comments and suggestions for Authors:
Point 1 The authors claim that: "The present study aimed to investigate the prevalence, molecular characteristics, and antibiotic susceptibility of polymyxin-resistant P. aeruginosa isolated frompatients in a Chinese tertiary teaching hospital and that In China, many studies reported themechanism of polymyxin...", so I have doubts about the novelty of the work. It should be clearly stated what is new about the work.
Response 1: Polymyxin resistance in China is uncommon. Most Chinese studies that have investigated mechanisms of polymyxin resistance have examined Enterobacterales, with few studies examining resistance in P. aeruginosa. Additionally, those studies that have analyzed polymyxin resistance in P. aeruginosa involve a small number of isolates collected across multiple centers across a wide region. Consequently, unlike our study they did not attempt to determine the prevalence of polymyxin resistance in a single tertiary hospital. We have improved the Introduction to better emphasis the novelty of our work, noting that this is also highlighted in the Discussion (Lines 102-106, Page 2 and Lines 659-668, Page 8).
Point 2
MALDI-TOF MS was used in the study, but no details are reported, there is no chromatogrametc. Line 86-87 MODLI-TOF MS should be written as MALDI-TOF MS.
Response 2: The MALDI-TOF MS system (BioMerieux, Missouri, France; software version 3.2) was used to identify polymyxin-resistant isolates of P. aeruginosa. This widely-used system generates standardized results on bacterial identification (without chromatograms) with high confidence (confidence value ≥99%). This point has been clarified in the manuscript and the typographical error corrected (Lines 114-115, Page 2).
Other changes:
We tried our best to improve the manuscript and made some changes in the manuscript. These changes will not influence the content and framework of the paper.
We appreciate for Editors/Reviewers’ warm work earnestly, and hope that the correction will meet with approval.
Once again, thank you very much for your comments and suggestions.
Reviewer 3 Report
The manuscript reports the low prevalence (8 isolates out of 362) of polymyxin-resistant P. aeruginosa in a Chinese teaching hospital and determined the genetic and drug-resistant phenotypes of the 8 resistant isolates. The analysis was based on molecular typing using PCR, multi-locus sequence typing (MLST) and whole-genome sequencing of the 8 polymyxin-resistant P. aeruginosa isolates and by antimicrobial susceptibility tests. The authors attempted to correlate the emergence of the polymyxin resistance to the use of this antibiotic. The results are interesting and deserve publication. However, I have some critical concerns:
1. Where the sequenced genomes assembled de novo? In this case which assembler tool was used?
2. The results derived from most of these tools listed in lines 105-114 are not presented in the manuscript (except for resistance genes indicated by CARD). Why list all these tools in the Methods?
3. Why the genome of Pseudomonas aeruginosa B136-33 (GenBank accession no. CP004061.1) was chosen to map the high-quality sequence reads for the 8 sequenced polymyxin-resistant P. aeruginosa? Why not the reference strains PA01 or PA14 genomes were used for the mapping?
4. Provide more information regarding Pseudomonas aeruginosa B136-33 such as ST, presence of resistance genes. Was this strain included in the SNP phylogeny?
5. I would suggest using isolates instead strains, particularly in the case of 362 isolates.
6. Why the presence of mcr gene was investigated by PCR? Isn´t the whole genome sequence reliable enough to demonstrate presence or absence of this gene?
7. It seems the 8 new genomes reported in this manuscript are not publicly available. No accession numbers were provided in current version of the manuscript. It is my understanding that data should be available in a publicly accessible repository according to MDPI policy.
8. Legend of Figure 2 should indicate that the phylogeny was based on SNPs.
9. Provide (as supplementary material) the list of accession numbers of the 385 complete genomes of P. aeruginosa from GenBank database used in the SNPs comparative analysis. Since there are thousands of P. aeruginosa publicly available, which criteria was used to select these 385 genomes?
10. Provide reference for the data mentioned on lines 217-218 (… a large number (> 50) of polymyxin-resistant strains have been detected including in Enterobacterales and A. bamannii.)
Author Response
Thank you for your letter and for the reviewers’ comments concerning our manuscript entitled “Prevalence and molecular characteristics of polymyxin-resistant Pseudomonas aeruginosa in a Chinese tertiary teaching hospital” (Submission ID: antibiotics-1748582). Those comments are all valuable and very helpful for revising and improving our paper, as well as the important guiding significance to our researches. We have studied comments carefully and have made correction which we hope meet with approval. The main corrections in the paper and the responds to the reviewer’s comments are as flowing:
All line and page numbers referred to below refer to the revised version of the manuscript. Please check the line numbers in the PDF file.
Comments and Suggestions for Authors
The manuscript reports the low prevalence (8 isolates out of 362) of polymyxin-resistant P. aeruginosa in a Chinese teaching hospital and determined the genetic and drug-resistant phenotypes of the 8 resistant isolates. The analysis was based on molecular typing using PCR, multi-locus sequence typing (MLST) and whole-genome sequencing of the 8 polymyxin-resistant P. aeruginosa isolates and by antimicrobial susceptibility tests. The authors attempted to correlate the emergence of the polymyxin resistance to the use of this antibiotic. The results are interesting and deserve publication. However, I have some critical concerns:
- Were the sequenced genomes assembled de novo? In this case which assembler tool was used?
Response 1: De novo genome assembly was conducted using A5-Miseq (v20160825) and SPAdes (v3.12.0), followed by base correction using Pilon. This has been clarified in the manuscript (Lines 259-261, Page 3).
- The results derived from most of these tools listed in lines 105-114 are not presented in the manuscript (except for resistance genes indicated by CARD). Why list all these tools in the Methods?
Response 2: We apologize for this error. The tools not directly related to results presented in the manuscript have been removed.
- Why the genome of Pseudomonas aeruginosa B136-33 (GenBank accession no. CP004061.1) was chosen to map the high-quality sequence reads for the 8 sequenced polymyxin-resistant aeruginosa? Why not the reference strains PA01 or PA14 genomes were used for the mapping?
Response 3: The genome of P. aeruginosa B136-33 was chosen because comparison of our genome assembly results with the NT library revealed the B136-33 genome was most closely match the genomes of the 3 polymyxin-resistant P. aeruginosa isolates tested. Consequently, this genome was selected as the reference for subsequent analysis. This has been clarified in the manuscript (Lines 267-270, Page 3).
- Provide more information regarding Pseudomonas aeruginosa B136-33 such as ST, presence of resistance genes. Was this strain included in the SNP phylogeny?
Response 4: P. aeruginosa B136-33 was ST-1024 and contains pmrA, pmrB, phoP and blaoxa-50 genes. It was included in the SNP phylogeny. This has been clarified in the manuscript (Lines 267-270, Page 3).
- I would suggest using isolates instead strains, particularly in the case of 362 isolates.
Response 5: Thank you for the suggestion. As request and where appropriate we have replaced ‘strains’ with ‘isolates’ throughout the revised manuscript.
- Why the presence of mcr gene was investigated by PCR? Isn´t the whole genome sequence reliable enough to demonstrate presence or absence of this gene?
Response 6: The mcr genes were detected using whole genome sequencing. We have deleted ‘during PCR screening’ to avoid any confusion (Line 466, Page 4).
- It seems the 8 new genomes reported in this manuscript are not publicly available. No accession numbers were provided in current version of the manuscript. It is my understanding that data should be available in a publicly accessible repository according to MDPI policy.
Response 7: The complete genome sequences of polymyxin-resistant P. aeruginosa isolates have been deposited in the NCBI BioProject repository under the accession numbers PRJNA846971. The reference to this has been included in the revised version of the manuscript (Lines 284-286, Page 3).
- Legend of Figure 2 should indicate that the phylogeny was based on SNPs.
Response 8: Thank you for the suggestion and this has been added to the legend of Figure 2 as requested (Line 651, Page8).
- Provide (as supplementary material) the list of accession numbers of the 385 complete genomes of P. aeruginosa from GenBank database used in the SNPs comparative analysis. Since there are thousands of P. aeruginosa publicly available, which criteria was used to select these 385 genomes?
Response 9: We have included the accession numbers of the 385 complete P. aeruginosa genomes in a Supplementary Material (Table X-pa.strains.complete.genome.info.xlsx). These P. aeruginosa strains were selected because they have complete genome sequences so that SNP calling would not be affected by assembly quality.
- Provide reference for the data mentioned on lines 217-218 (… a large number (> 50) of polymyxin-resistant strains have been detected including in Enterobacterales and A. baumannii.)
Response 10: These data are currently unpublished. We have deleted reference to these data from the discussion (Line 662, Page 8).
Other changes:
We tried our best to improve the manuscript and made some changes in the manuscript. These changes will not influence the content and framework of the paper.
We appreciate for Editors/Reviewers’ warm work earnestly, and hope that the correction will meet with approval.
Once again, thank you very much for your comments and suggestions.
Round 2
Reviewer 2 Report
I suggest publication of the work in the present form.